# How Knowledge Popularity Influences and Enhances LLM Knowledge Boundary Perception

## Abstract

Large language models (LLMs) often fail to recognize their knowledge boundaries, producing confident yet incorrect answers. In this paper, we investigate how knowledge popularity affects LLMs' ability to perceive their knowledge boundaries. Focusing on entity-centric factual question answering (QA), we quantify knowledge popularity from three perspectives: the popularity of entities in the question, the popularity of entities in the answer, and relation popularity, defined as their co-occurrence frequency. Experiments on three representative datasets containing knowledge with varying popularity show that LLMs exhibit better QA performance, higher confidence, and more accurate perception on more popular knowledge, with relation popularity having the strongest correlation. Cause knowledge popularity shows strong correlation with LLMs' QA performance, we propose to leverage these signals for confidence calibration. This improves the accuracy of answer correctness prediction by an average of 5.24% across all models and datasets. Furthermore, we explore prompting LLMs to estimate popularity without external corpora, which yields a viable alternative.

## 1 Introduction

Large language models (LLMs) (Achiam et al., 2023; Yang et al., 2024; Dubey et al., 2024) often hallucinate—producing fluent but factually incorrect answers which is unacceptable in safety-critic domains such as healthcare. Accurately identifying when LLMs produce correct answers not only helps determine when to trust their outputs, but also enables adaptive retrieval-augmented generation (RAG)—performing retrieval only when they do not know the answer—thereby enhancing both the effectiveness and efficiency of RAG (Ni et al., 2024a). A reliable model should have a clear perception of its knowledge boundaries—knowing what it knows and what it does not. This requires its confidence in an answer, reflected in the generation probability, to align with the actual likelihood of the answer being correct (Jiang et al., 2021). While many studies have examined LLMs' perception level of their knowledge boundaries and found that they tend to be overconfident (Lin et al., 2022; Tian et al., 2023), the underlying factors influencing the perception remain poorly understood.

A natural hypothesis is that a model's perception level can be influenced by the popularity of the knowledge—i.e., how frequently the model has encountered it during training. When asked about popular knowledge, the model may be more likely to respond both correctly and confidently. Prior work (Mallen et al., 2023) has shown that LLMs achieve better QA performance on more popular questions. This raises a key question: how do the model's confidence and its alignment with QA performance vary with knowledge popularity?

To investigate this, we focus on entity-centric factual QA (Mallen et al., 2023; Yuksekgonul et al., 2023) where both the question and the answer contain an entity because this enables us to quantify the popularity of knowledge based on entities. Specifically, we assess knowledge popularity from the following three perspectives: 1) *Question popularity $Pop_Q$*: popularity of the entity in the question. 2) *Ground-truth answer popularity $Pop_{GT}$*: popularity of the entity in the ground-truth answer. 3) *Ground-truth relation popularity $RPop_{GT}$*: the co-occurrence frequency of the question and ground-truth entities. Higher entity popularity suggests more accurate entity representations. Rela-

tion popularity directly influences the model's ability to comprehend associations between entities, but is costly to collect.

Since ground-truth answers are unavailable in real-world scenarios, we also examine model-generated answers. Specifically, we investigate whether the popularity of the generated answer ($Pop_{Ge}$) and the relation popularity between the generated and question entities ($RPop_{Ge}$) reflect the model's QA performance, confidence, and perception level. We focus in particular on their correlation with QA performance, as a strong correlation could allow these signals to be used for calibrating the model's confidence.

We conduct experiments on three entity-centric factual QA datasets—Movies, Songs, and Basketball—constructed from Wikidata knowledge triplets by Yuksekgonul et al. (2023). Some question examples can be seen in Figure 1. We quantify entity popularity by the number of Wikidata language editions in which an entity appears. Relation popularity is measured by the number of Wikipedia documents where both entities are mentioned together. We use two representative open-source models—LLaMA3-8B-Instruct (Dubey et al., 2024) and Qwen2-7B-Instruct (Yang et al., 2024)—as well as the black-box model ChatGPT (Achiam et al., 2023).

Results on $Pop_Q$, $Pop_{GT}$, and $RPop_{GT}$ show that *LLMs demonstrate better QA performance, higher confidence, and more accurate perception of their knowledge boundaries on more popular knowledge.* Although LLMs are generally overconfident, the extent of overconfidence diminishes as knowledge popularity increases, since QA performance improves more rapidly than confidence. Among the three popularity measures, $RPop_{GT}$ shows the strongest correlation with QA accuracy, confidence, and perception level most of the cases. Interestingly, question popularity correlates more strongly with confidence than with QA performance, implying that LLMs may become overconfident simply due to familiarity with the question.

Regarding generated answers, $RPop_{Ge}$ shows a strong positive correlation with QA performance, confidence, and perception level, while $Pop_{Ge}$ exhibits a weaker correlation. Notably, $RPop_{Ge}$ shows even stronger correlation with QA performance than $RPop_{GT}$, while $Pop_{Ge}$ correlates more weakly than $Pop_{GT}$. We further analyze the reason and reveal that *when LLMs make errors, they tend to generate more popular entities that co-occur less frequently with the question entity compared to ground-truth answers*, indicating a tendency toward over-generalization. This is consistent with the findings of Zhang et al. (2024b).

Based on these findings, we propose to leverage popularity features (i.e., $Pop_Q$, $Pop_{Ge}$, and $RPop_{Ge}$) to calibrate confidence which aims to improve the effectiveness of confidence in predicting answer correctness. Given that computing knowledge popularity requires access to external corpora and incurs additional collection costs, we also investigate prompting the model to estimate popularity on its own. Results show that $Pop_Q$ and $Pop_{Ge}$ provide modest gains in calibration. In contrast, $RPop_{Ge}$ provides substantial gains. *Combining all these three types of popularity yields the best calibration performance, boosting answer correctness prediction by an average of 5.24% across all models and datasets.* Moreover, leveraging model-estimated popularity also performs well for confidence calibration. The choice between external corpora and self-estimation ultimately hinges on the trade-off between performance and efficiency.

## 2 RELATED WORK

Existing research on model knowledge boundary perception focuses on assessing model confidence and can be mainly classified into four categories.

**Probabilistic Confidence.** This line of research treats the generation probability of the answer as the confidence of the model (Guo et al., 2017; Desai & Durrett, 2020; Jiang et al., 2021; Kadavath et al., 2022; Si et al., 2022; Kuhn et al., 2023). Guo et al. (2017) examined early neural networks (e.g., ResNet (He et al., 2016)) and found them to be overconfident, proposing temperature scaling as a remedy. Later, Desai & Durrett (2020) showed that BERT-style models tend to be relatively well-calibrated, while Jiang et al. (2021) found that pre-trained language models such as T5 (Raffel et al., 2020) remained overconfident. More recent work has turned to LLMs, with studies showing that they, too, exhibit overconfidence (Si et al., 2022; Lin et al., 2022; Tian et al., 2023).

**Verbalized Confidence.** LLMs have been shown to express their confidence verbally (Lin et al., 2022; Yin et al., 2023; Tian et al., 2023; Xiong et al., 2023; Yang et al., 2023; Ni et al., 2024a). Some studies (Yin et al., 2023; Ni et al., 2024a) found that LLMs often fail to recognize their knowledge limitations verbally and tend to be overconfident. Xiong et al. (2023) systematically studied black-box approaches for estimating LLM confidence. Beyond prompting-based methods, some studies aim to train LLMs to verbalize more accurate confidence (Lin et al., 2022; Yang et al., 2023; Zhang et al., 2024a).

**Consistency-based Confidence.** If the model is confident in its answer, it should maintain consistency across multiple generations. Recent studies have used self-consistency across generations as a proxy for LLM confidence (Manakul et al., 2023; Kuhn et al., 2023). Zhang et al. (2023) extended this by evaluating the consistency of answers across multiple semantically equivalent inputs and across different models. Ding et al. (2024) further adapted this approach to the multilingual setting.

**Confidence Estimation via LLM Internal States.** LLMs' internal states have shown to be effective in evaluating the factuality of their self-generated content (Su et al., 2024; Chen et al., 2024; Wang et al., 2024; Ni et al., 2025). Specifically, Su et al. (2024) and Chen et al. (2024) focused on internal states after generation, Wang et al. (2024) examined those before generation, and Ni et al. (2025) explored leveraging LLMs' internal states to enhance their perception of knowledge boundaries from efficiency and risk perspectives.

We focus on probabilistic confidence for the following reasons: 1) Both the model's generation probabilities and its knowledge acquisition arise from the same training objective, and are expected to align with each other. 2) Models without specialized training often struggle to verbalize confidence accurately (Ni et al., 2024b); consistency-based methods require multiple generations and incur high inference costs; and internal-state-based approaches require access to hidden representations and additional training. In contrast, probabilistic confidence is readily accessible and has been shown to perform well, especially when answers are short (Ding et al., 2024).

# 3 TASK DESCRIPTION

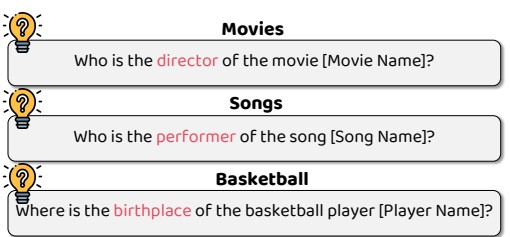

Figure 1: Question examples for each dataset.

**Entity-Centric QA.** We focus on entity-centric knowledge because it allows us to measure knowledge popularity through entities. In entity-centric QA, questions and answers are derived from knowledge triples in the form of (subject, relation, object), where the question queries the relation of a given subject, and the model is expected to generate the corresponding object. Examples of knowledge triples are provided in Table 1, with their transformed question forms shown in Figure 1.

**LLM Knowledge Boundary Perception.** The model's perception of its knowledge boundaries is evaluated by the alignment between its confidence and actual QA performance. QA performance is measured by whether the generated answer contains the ground-truth answer, and confidence is reflected in generation probability of the answer tokens (see Section § 2). Specifically, for a question $q$ and a model $M$, the confidence $c$ is computed as:

$$c = \frac{1}{T} \sum_{i=1}^{T} P(g_i|g_{<i}),$$

(1)

where $\{g_1, \ldots, g_T\}$ is the generated tokens.

# 4 HOW DOES KNOWLEDGE POPULARITY AFFECT LLMS' PERCEPTION LEVEL?

In this section, we investigate how $\text{Pop}_Q$, $\text{Pop}_{GT}$, and $\text{RPop}_{GT}$ influence LLMs' QA performance, confidence, and perception level.

Table 1: Data size for each dataset, along with the corresponding subject, relation, and object types.

| Datasets | Count | Subject | Relation | Object |
|---|---|---|---|---|
| Movies | 10,964 | Movie | Directed by | Director |
| Songs | 2,157 | Song | Performed by | Performer |
| Basketball | 13,309 | Player | Birthplace | City |

Table 2: Definitions of notations about knowledge popularity where pop. means popularity.

| Notation | Definition |
|---|---|
| $\text{Pop}_Q$ | Popularity of entities in the question |
| $\text{Pop}_{GT}$ | Popularity of entities in the ground-truth answer |
| $\text{RPop}_{GT}$ | Relation pop. between question and ground-truth entities |
| $\text{Pop}_{Ge}$ | Popularity of entities in the generated answer |
| $\text{RPop}_{Ge}$ | Relation pop. between question and generated entities |

## 4.1 EXPERIMENTAL SETUP

**Datasets.** Yuksekgonul et al. (2023) constructed entity-centric QA datasets based on Wikidata[1], using the number of sitelinks on a page as a proxy for entity popularity. They showed that this measure strongly correlates with an entity's frequency in the training data. Building on this, we conduct experiments on their datasets to ensure reliable entity popularity measurement. We select three representative datasets—Movies, Song, and Basketball—because they exhibit clear differences in knowledge popularity. Specifically, question popularity ranks as Movies ¿ Songs ¿ Basketball, while ground-truth answer popularity follows Movies ¡ Songs ¡ Basketball. Table 1 lists the knowledge triplets and data counts for each dataset, and Figure 1 presents example questions. We apply data filtering to ensure reliable results, as detailed in Section §A.

**Entity Popularity.** Following Mallen et al. (2023); Yuksekgonul et al. (2023), we define the popularity of an entity by the number of sitelinks it has—i.e., the number of Wikipedia pages in different languages that link to it.

**Relation Popularity.** As Wikipedia is the primary high-quality source for Wikidata, we estimate relation popularity based on Wikipedia content. Specifically, for each entity pair, we measure relation popularity by counting the number of documents in the Wikipedia dump[2] in which both entities co-occur. This reflects relation popularity in the model's training data, as it shows a strong correlation with QA performance (see Table 3).

**LLMs.** We conduct experiments on three representative LLMs: two open-source models, Llama3-8B-Instruct (Dubey et al., 2024) and Qwen2-7B-Instruct (Yang et al., 2024), as well as a black-box model, ChatGPT (i.e., GPT-3.5-Turbo-1106) (Achiam et al., 2023).

**Answer Generation.** For all the models, we use greedy search, selecting the token with the highest probability at each generation step. An example can be seen in Figure 20.

**Metrics.** For each question $q_i$, we measure answer correctness using accuracy $acc_i$, where the generated answer is considered correct if it contains the ground-truth answer. The model's confidence $c_i$ is defined as the generation probability of the answer, as described in Section §3. Alignment is

---

[1] https://query.wikidata.org/sparql
[2] https://huggingface.co/datasets/wikimedia/wikipedia

Table 3: LLMs' QA performance, confidence, alignment and the correlations between knowledge popularity and accuracy, confidence, and alignment across different datasets.

| Datasets | Models | Acc. | Accuracy | | | Conf. | Confidence | | | Align. | Alignment | | |
|---|---|---|---|---|---|---|---|---|---|---|---|---|---|
| | | | $Pop_Q$ | $Pop_{GT}$ | $RPop_{GT}$ | | $Pop_Q$ | $Pop_{GT}$ | $RPop_{GT}$ | | $Pop_Q$ | $Pop_{GT}$ | $RPop_{GT}$ |
| Movies | Llama3 | 72.65 | 0.317 | 0.220 | **0.357** | 90.68 | 0.404 | 0.367 | **0.509** | 75.50 | 0.404 | 0.347 | **0.501** |
| | Qwen2 | 42.85 | 0.433 | 0.299 | **0.494** | 82.32 | 0.413 | 0.371 | **0.507** | 53.63 | 0.386 | 0.279 | **0.440** |
| | ChatGPT | 94.78 | **0.134** | 0.069 | 0.130 | 98.80 | 0.210 | 0.230 | **0.280** | 94.85 | 0.211 | 0.228 | **0.279** |
| Songs | Llama3 | 38.97 | 0.277 | 0.164 | **0.517** | 79.74 | 0.369 | 0.210 | **0.502** | 53.04 | 0.182 | 0.093 | **0.361** |
| | Qwen2 | 25.82 | 0.362 | 0.255 | **0.541** | 78.00 | 0.300 | 0.200 | **0.345** | 42.97 | 0.230 | 0.180 | **0.392** |
| | ChatGPT | 73.36 | 0.171 | 0.266 | **0.399** | 94.84 | 0.249 | 0.295 | **0.381** | 75.28 | 0.232 | 0.340 | **0.399** |
| Basketball | Llama3 | 13.37 | 0.118 | **0.293** | 0.231 | 60.09 | **0.173** | 0.063 | 0.055 | 46.21 | -0.052 | **0.104** | 0.097 |
| | Qwen2 | 9.90 | 0.014 | **0.348** | 0.151 | 74.76 | **0.151** | 0.076 | 0.009 | 32.35 | 0.126 | **0.189** | 0.105 |
| | ChatGPT | 34.89 | 0.288 | 0.215 | **0.353** | 79.06 | **0.351** | 0.054 | 0.270 | 50.43 | 0.201 | 0.164 | **0.303** |

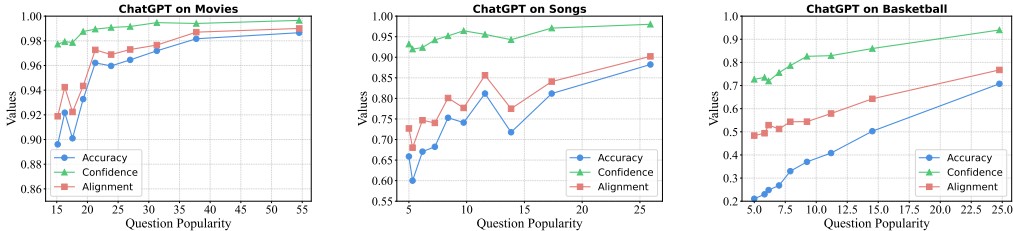

Figure 2: QA performance, confidence, and alignment of ChatGPT across question popularity.

then computed as $1 - |acc_i - c_i|$. To examine the relationship between knowledge popularity and QA performance, confidence, and alignment, we use Spearman correlation coefficients (Hauke & Kossowski, 2011), which range from -1 to 1. The sign indicates the direction of the correlation, while the absolute value reflects its strength.

## 4.2 RESULTS AND ANALYSIS

LLMs' QA performance, confidence, and perception levels across different datasets, along with the Spearman correlation coefficients between knowledge popularity and accuracy, confidence, and alignment are shown in Table 3. We observe that:

**1) LLMs achieve better QA performance and higher confidence on more popular knowledge.**
All three types of popularity are positively correlated with both QA performance and confidence. For QA performance and confidence, we conduct the following analyses respectively.

For QA performance, relation popularity generally shows the strongest correlation, as expected, while question popularity exhibits a stronger correlation than answer popularity in most cases. This suggests that learning through co-occurrence is especially effective for acquiring knowledge, and that familiarity with the question contributes more to answering correctly than familiarity with the answer. However, on the Basketball dataset, answer popularity shows the highest correlation for both LLaMA3 and Qwen2. This dataset is challenging because both the question and relation have low popularity. The models are often unfamiliar with the question entity and generate a popular city name instead—a behavior consistent with knowledge overshadowing (Zhang et al., 2024b). ChatGPT does not exhibit this pattern, likely due to a stronger mastery of the relevant knowledge.

For confidence, question and relation popularity are strongly correlated in most cases, while answer popularity has a weaker impact. Notably, question popularity consistently correlates strongly with confidence and, in 7 of 9 cases, more than with QA performance. This suggests that LLMs may become more confident simply because familiarity with the question, even if they do not know the answer. On the Basketball dataset, confidence shows little correlation with answer popularity across all three models. We hypothesize that the models are generally familiar with city names, and thus do not exhibit higher confidence for samples with more common answers.

**2) LLMs better perceive their knowledge boundaries on more popular knowledge.** To better understand this, we analyze how the gap between confidence and QA performance changes with

Table 4: Correlations between LLMs' QA performance and $Pop_Q$, $Pop_{Ge}$, and $RPop_{Ge}$.

| Datasets | Models | $Pop_Q$ | $Pop_{Ge}$ | $RPop_{Ge}$ |
|---|---|---|---|---|
| Movies | Llama3 | 0.317 | 0.100 | **0.637** |
|  | Qwen2 | 0.433 | 0.087 | **0.756** |
|  | ChatGPT | 0.134 | 0.083 | **0.208** |
| Songs | Llama3 | 0.277 | 0.257 | **0.621** |
|  | Qwen2 | 0.362 | 0.188 | **0.666** |
|  | ChatGPT | 0.171 | 0.218 | **0.351** |
| Basketball | Llama3 | 0.118 | 0.116 | **0.245** |
|  | Qwen2 | 0.014 | **0.116** | 0.106 |
|  | ChatGPT | 0.288 | -0.164 | **0.293** |

increasing knowledge popularity. Due to space constraints, we just present this gap for ChatGPT as question popularity increases, shown in Figure 2. We observe that although LLMs are consistently overconfident, their QA performance improves more rapidly than confidence as question popularity increases, thereby narrowing the gap. Results for other models, as well as analyses based on other popularity, are included in the Appendix and exhibit similar trends. As shown in Table 3, among the three types of popularity, relation popularity typically shows the strongest correlation.

# 5 ANALYSIS OF MODEL-GENERATED ANSWERS

In real-world scenarios, ground-truth entities are often unavailable. This motivates us to investigate whether the popularity of model-generated entities—along with their relational popularity with the question entity—correlates with the model's QA performance, confidence, and perception level. We focus particularly on the relationship between popularity and QA performance, as a strong correlation could enable us to leverage these signals for confidence calibration. The experimental settings are the same as those in Section § 4.

## 5.1 RESULTS AND ANALYSIS

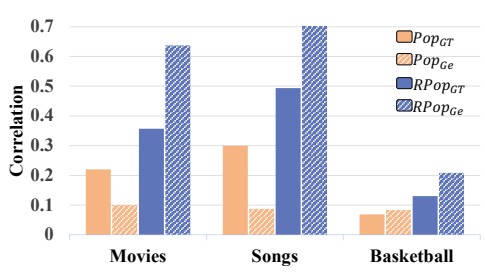

Figure 3: Comparison of the correlation between ChatGPT's QA performance and ground-truth vs. generated answers: $Pop_{GT}$ vs. $Pop_{Ge}$, and $RPop_{GT}$ vs. $RPop_{Ge}$.

Table 4 shows the Spearman correlation coefficients between LLMs' QA performance and knowledge popularity based on model-generated entities. Due to space constraints, results on LLMs' confidence and perception levels are provided in Table 7 in the Appendix. We observe the following.

The popularity of generated entities ($Pop_{Ge}$) and their co-occurrence with question entities ($RPop_{Ge}$) positively correlate with LLMs' QA performance, confidence, and perception level in most cases. $RPop_{Ge}$ typically shows the strongest correlation, outperforming both $Pop_Q$ and $Pop_{Ge}$. In contrast, $Pop_{Ge}$ often exhibits the weakest correlation. These findings are similar to the results based on ground-truth entities, as discussed in Section § 4.

$Pop_{Ge}$ shows a weaker correlation with QA performance compared to $Pop_{GT}$ while $RPop_{Ge}$ exhibits a comparable or even stronger correlation than $RPop_{GT}$. We present the comparison for ChatGPT in Figure 3, while results for other models can be obtained by comparing Table 3 and Table 4. To better understand this, we perform a more detailed comparison between model-generated answers and ground-truth answers. We only focus on cases where the model makes mistakes since the generated answer matches the ground-truth answer otherwise and analyze in Section § 5.2.

## 5.2 WHAT DO LLMs GENERATE WHEN THEY HALLUCINATE?

We focus on the differences in popularity between model-generated answers and ground-truth answers when the model makes errors (See Figure 4), as well as the differences in their co-occurrence frequency with the question entity (See Figure 5).

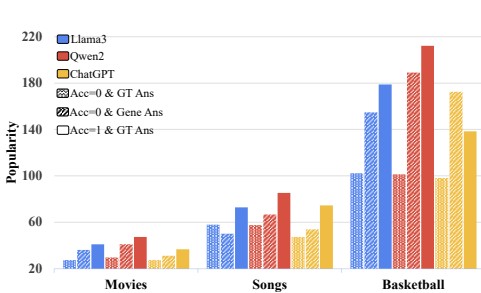

Figure 4: $Pop_{GT}$, $Pop_{Ge}$ in incorrectly answered samples and $Pop_{GT}$ (also $Pop_{Ge}$) in correctly answered samples.

**LLMs tend to generate entities that are more common and less frequently co-occur with the question entities than ground-truth answers when they hallucinate.** As shown in Figure 4, for incorrectly answered samples, the generated entities are often more popular than the ground-truth entities. So the popularity gap between correct and incorrect answers is smaller for generated entities than for ground-truth entities, leading to a weaker correlation between $Pop_{Ge}$ and QA performance. This may be because the model tends to overgeneralize—once it learns high-frequency answers, it tends to use them in many similar contexts.

In incorrectly answered samples, model-generated answers typically co-occur less frequently with question entities compared to ground-truth answers, as shown in Figure 5. As a result, the correlation between $RPop_{Ge}$ and QA performance is stronger than that of $RPop_{GT}$, because the difference in $RPop_{Ge}$ between correct and incorrect samples is greater than that of $RPop_{GT}$.

## 6 CONFIDENCE CALIBRATION WITH KNOWLEDGE POPULARITY

Given that $Pop_Q$, $Pop_{Ge}$, and $RPop_{Ge}$—especially $RPop_{Ge}$—are strongly correlated with QA performance, we propose to use these signals for confidence calibration, i.e., improving the effectiveness of confidence in predicting answer correctness. Since obtaining these signals typically requires external corpora and incurs additional costs, we also explore whether LLMs can assess knowledge familiarity on their own.

### 6.1 KNOWLEDGE POPULARITY ACQUISITION

**Corpora-based Popularity.** As outlined in Section § 3, we get popularity from external corpora.

Figure 5: Proportion of incorrectly answered samples where $RPop_{Ge}$ is less than $RPop_{GT}$.

**Model-generated Popularity.** To eliminate reliance on external corpora and reduce the overhead of collecting popularity, we investigate whether LLMs can self-assess their familiarity with a given the entity or the relation. Familiarity is measured on a 10-point scale, where 1 denotes the lowest and 10 the highest level. The model is asked to provide its familiarity score accordingly. We provide the model with varying numbers of corpora-based popularity examples to examine whether supplying such examples helps the model produce more accurate familiarity. We present examples under both zero-shot and few-shot settings and all these prompts can be found in Section § F in the Appendix.

### 6.2 BASELINES

We use representative confidence estimation methods that do not require access to model parameters as our baselines.

- **Verbalized Confidence (Verb)** (Yin et al., 2023) instructs the model to verbally assess whether it can answer the question correctly. The prompt can be seen in Figure 21 in the appendix.

- **Self-consistency (Consis)** (Manakul et al., 2023) estimates the model's confidence by measuring the semantic consistency of multiple sampled answers. The core idea is that if the model knows the correct answer, multiple sampled answers should be semantically consistent. For each question, we sample 10 additional responses with a temperature of 1.0. For ChatGPT, due to cost constraints, we limit sampling to 3 additional responses. Semantic consistency is assessed using Qwen2.5-32B-Instruct (Yang et al., 2024).
- **Probabilistic Confidence (PC)** (Kumar et al., 2024) takes generation probabilities for the tokens in the answer as the model's confidence. This is the confidence we have been using throughout the paper, and the one we aim to calibrate. Details can be seen in Section § 3.

### 6.3 DATA CONSTRUCTION

For each sample in the datasets, we construct a data pair $\{x, y\}$, where $y$ is a binary correctness label: $y = 1$ if the generated answer contains the ground-truth answer, and $y = 0$ otherwise. To study the effect of each type of popularity on confidence calibration, we construct $x$ using the following features: 1) **PC+Pop$_{\text{Q}}$**, 2) **PC + Pop$_{\text{Ge}}$**, 3) **PC + RPop$_{\text{Ge}}$**, and 4) **PC + ALL** — PC and all these three types of popularity. We also use each type of popularity itself as $x$ to study its effectiveness in answer correctness prediction.

### 6.4 ANSWER CORRECTNESS PREDICTION

Based on the constructed $x$, we predict a binary judgment on correctness, as shown below.

$$\hat{y} = \mathcal{E}(x), \tag{2}$$

where $\mathcal{E}$ represents the binary classification function and $\hat{y}$ means the predicted correctness (i.e., 1 for correct and 0 for incorrect).

**Single-feature Prediction.** For the setting where $x$ contains only a single feature, we select a threshold $\lambda$ that maximizes prediction accuracy on the training set, and apply this threshold to perform binary classification on the test data. This can be formualted as:

$$\hat{y} = \begin{cases} 1 & \text{if } x > \lambda, \\ 0 & \text{otherwise}, \end{cases} \tag{3}$$

**Multi-feature Prediction.** For the setting where $x$ contains multiple features, we perform binary classification using a lightweight MLP network, as defined below:

$$P(\hat{y} = 1) = \sigma\left(\text{MLP}\left(x\right)\right), \tag{4}$$

where $\sigma$ refers to the softmax function, $x \in \mathcal{R}^{d \times h}$ represents the input features, $d$ is the count of input features (e.g., $d$=2 for PC+Pop$_{\text{Q}}$) and $h$ means the model's hidden dimension. We use a 3-layer MLP with 64, 32, and 2 neurons in each layer, respectively. The activation function in MLP is ReLU. We employ cross-entropy loss as the training objective:

$$\mathcal{L}_{\text{CE}} = -\sum_{i=1}^{N} y_i \log(P_i) + (1 - y_i) \log(1 - P_i), \tag{5}$$

where $y_i$ is the ground-truth correctness for the $i$-th training sample, $N$ is the count of training samples, and $P_i$ denotes $P(\hat{y}_i = 1)$. Detailed training parameters can be found in Section § D.

**Metrics.** We use answer correctness prediction accuracy as the metric. To reduce the impact of randomness, all our reported results are the averages obtained from three random seeds: (0, 42, 100).

**Datasets and LLMs.** We use the same data and LLMs as in Section § 4.1. We randomly split each dataset into two equal parts for training and testing and select the checkpoint with the highest prediction accuracy on the training set. Detailed settings can be found in Section § D in the Appendix.

Table 5: Accuracy of answer correctness prediction. Bold denotes the highest score in either corpora-based or self-generated knowledge popularity. Self-generated knowledge popularity is obtained under the zero-shot setting.

| Features | Movies | | | Songs | | | Basketball | | | Avg. |
|---|---|---|---|---|---|---|---|---|---|---|
| | Llama3 | Qwen2 | ChatGPT | Llama3 | Qwen2 | ChatGPT | Llama3 | Qwen2 | ChatGPT | |
| | | | | *Baselines* | | | | | | |
| Verb | 65.58 | 45.93 | 83.41 | 40.22 | 29.58 | 69.25 | 51.58 | 50.49 | 48.89 | 53.88 |
| Consis | 82.21 | 74.61 | 96.00 | 77.62 | 86.31 | 83.72 | 53.76 | 52.10 | 77.77 | 76.01 |
| PC | 83.20 | 79.77 | 95.95 | 75.20 | 83.02 | 79.11 | 65.49 | 66.36 | 77.87 | 78.44 |
| | | | | *Corpora-based Knowledge Popularity* | | | | | | |
| $Pop_Q$ | 71.68 | 70.62 | 88.24 | 66.35 | 76.84 | 68.00 | 56.25 | 50.90 | 69.32 | 68.69 |
| $Pop_{Ge}$ | 73.09 | 58.86 | 94.22 | 63.38 | 74.57 | 74.41 | 60.54 | 60.33 | 64.86 | 69.36 |
| $RPop_{Ge}$ | 89.66 | 87.92 | 96.03 | **82.71** | 89.59 | 81.46 | 67.03 | 59.64 | 64.74 | 79.86 |
| $PC+Pop_Q$ | 83.57 | 81.36 | 95.97 | 76.60 | 84.58 | 79.11 | 65.93 | 66.95 | 78.39 | 79.16 |
| $PC+Pop_{Ge}$ | 84.46 | 80.49 | 95.58 | 76.68 | 83.57 | 80.12 | 69.04 | 68.64 | **78.62** | 79.69 |
| $PC+RPop_{Ge}$ | 90.93 | 88.58 | 96.13 | 80.21 | **90.46** | **84.04** | 71.93 | 66.33 | 78.10 | 82.97 |
| PC+ALL | **93.32** | **92.47** | **96.37** | 81.46 | 88.11 | 82.71 | **71.93** | 68.18 | 78.59 | **83.68** |
| | | | | *Self-generated Knowledge Popularity* | | | | | | |
| $PC+Pop_Q$ | 83.91 | 80.60 | 95.87 | 77.85 | 84.82 | 79.50 | 65.30 | 67.31 | 78.43 | 79.29 |
| $PC+Pop_{Ge}$ | 84.02 | 80.24 | 95.59 | 75.20 | 83.02 | 78.56 | **68.40** | 67.49 | 78.21 | 78.97 |
| $PC+RPop_{Ge}$ | 85.30 | 80.20 | 95.80 | **79.65** | 84.04 | 79.81 | 66.17 | 67.90 | 77.59 | 79.61 |
| PC+ALL | **85.95** | **81.40** | 95.84 | 78.87 | **86.07** | **80.05** | 67.69 | **68.08** | **78.70** | **80.29** |

## 6.5 RESULTS AND ANALYSIS

**Results on corpora-based knowledge popularity.** Results based on knowledge popularity from external corpora is shown in the upper half of Table 5. We observe that: 1) Compared to the model's confidence, $RPop_{Ge}$ more accurately reflects answer correctness, outperforming all baselines in 6 out of 9 cases. In contrast, $Pop_Q$ and $Pop_{Ge}$ individually show limited effectiveness in predicting correctness. 2) **All three types of popularity contribute to calibrating the model's confidence, with their combination yielding the most effective results.** In most cases, augmenting PC with each type of popularity improves upon PC, with $PC+RPop_{Ge}$ achieving the highest average accuracy among them. Notably, combining all three types leads to the most effective calibration, consistently outperforming PC and yielding an average accuracy improvement of 5.24% across diverse datasets and models. Further analysis and case studies are provided in Section § E.

**Results on model-generated knowledge popularity.** The prediction accuracy based on model self-generated knowledge popularity under the zero-shot setting can be found in the lower half of Table 5. It show that: 1) All three types of self-generated popularity contribute to confidence calibration. On average, all three signals can calibrate PC, and their combination achieves the best calibration effect, obtaining the optimal value in 6 out of 9 cases. However, the model's self-generated signals yield weaker calibration effects compared to corpus-based knowledge popularity. The choice between corpus-based popularity and self-generated popularity depends on the trade-off between effectiveness and efficiency. 2) LLMs can not estimate popularity better with few-shot learning compared to zero-shot. Detailed analysis can be found Section § C in the Appendix.

## 7 CONCLUSION

In this paper, we investigate how knowledge popularity—measured through entity and relation popularity—affects LLMs' QA performance, confidence, and perception of their knowledge boundaries, and explore its utility for confidence calibration. We find that LLMs perform better, express higher confidence, and demonstrate more accurate perception on more popular knowledge, with relation popularity having the strongest influence. We further show that the popularity and co-occurrence of model-generated answers also positively correlate with QA accuracy. Leveraging these popularity signals for confidence calibration yields an average 5.24% improvement in predicting answer correctness. To reduce reliance on external corpora, we also demonstrate that model-estimated popularity can serve as a viable alternative, offering a practical trade-off between performance and efficiency.

## ETHICS STATEMENT

We approach ethics with great care. In this paper, all the datasets and models we use are open-source. Our analysis of knowledge popularity does not introduce any harmful information. Moreover, our proposed method can help accurately determine whether the model's answer is trustworthy, preventing users from being misled by incorrect responses.

## REPRODUCIBILITY STATEMENT

All datasets used in this paper are publicly available, and the popularity signals were constructed based on these datasets. The detailed procedure is described in Section 4.1. The three LLMs employed in this study are widely adopted models. Furthermore, for training, we only used a lightweight MLP network, which requires minimal computational resources. Furthermore, all prompts used in this paper are provided in Section **??**.

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

# A  DATA FILTERING

The model's generated response may be empty or fail to find a corresponding entity in Wikidata. To ensure comparability of results across different models on the same dataset, we filter out data where any model's generation is empty or the generated entity cannot be found in Wikidata. Additionally, for the Movies and Songs datasets, we filter out cases where the question entity, ground truth entity, or model-generated entity appears in more than 6,000 documents. This is because entities in these two datasets typically do not appear in more than 6,000 documents, and those that exceed this threshold often introduce noise. For example, "Queen" appears more than 6,000 times but is not exclusively used as a band name. We filter these cases to obtain an accurate co-occurrence counts. After filtering, the remaining data sizes for the Movies, Songs, and Basketball datasets are 8,184, 852, and 13,136, respectively.

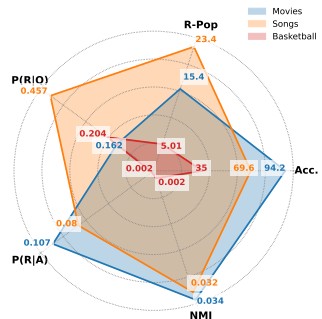

Figure 6: QA performance and NMI calculated based on ChatGPT. R-Pop means relation popularity, where $P(R|Q)$ and $P(R|A)$ denote the co-occurrence proportion of question and answer entities relative to their individual occurrences in documents.

# B  ANALYSIS ON RELATIONSHIP STRENGTH

We hypothesize that the strength of the relationship between entities may also influence the model's learning. Specifically, when the subject and object frequently co-occur but are also commonly associated with other entities, the model may struggle to learn their specific relationship. We use normalized mutual information to quantify relationship strength and find that stronger relationship contributes to better QA performance. Normalized mutual information is computed as:

$$\text{NMI}(X, Y) = \frac{I(X; Y)}{\sqrt{H(X)H(Y)}}, \tag{6}$$

where $I(X, Y)$ is defined as:

$$I(X, Y) = \sum_{i=1}^{n} \sum_{j=1}^{m} P(x_i, y_j) \log \frac{P(x_i, y_j)}{P(x_i)P(y_j)}, \tag{7}$$

and $H(X)$ and $H(Y)$ serve as regularization terms to mitigate the influence of the sizes of $n$ and $m$, as well as the magnitude of probability values. Their formulations are:

$$H(X) = -\sum_{i=1}^{n} P(x_i) \log P(x_i), \tag{8}$$

$$H(Y) = -\sum_{j=1}^{m} P(y_j) \log P(y_j). \tag{9}$$

Specifically, for a dataset converted from knowledge triplets $D = \{s_i, r, o_i\}_{i=1}^{n}$, we define $X = \{s_1, \ldots, s_n\}$ and $Y = \{o_1, \ldots, o_n\}$. We estimate $P(s_i)$, $P(o_j)$, and $P(s_i, o_j)$ using a Wikipedia dump of $d$ documents, where $P(s_i)$ and $P(o_j)$ are the proportions of documents containing $s_i$ and

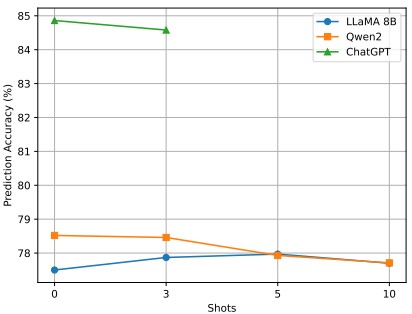

Figure 7: The prediction accuracy obtained by performing confidence calibration using knowledge popularity generated from different numbers of examples. Each point represents the average prediction accuracy of the model across three datasets.

$o_j$, respectively, and $P(s_i, o_j)$ is the proportion containing both. If $i \neq j$, we set $P(s_i, o_j) = 0$, as we focus only on the relationship between $s_i$ and $o_i$.

The results based on ChatGPT are shown in Figure 6. We observe that, compared to the movies dataset, the question entity co-occurs more frequently with the ground-truth entity in the songs dataset, yet the model's QA performance is lower. This can be attributed to the lower NMI in the songs dataset, driven by a low $P(R|A)$. This indicates that, besides the question entity, the answer entity also co-occurs with many other entities through various relations. This may interfere with the model's memory of the relationship between the question entity and the answer entity.

## C  Effects of Few-Shot Learning on Popularity Generation

**Examples selection.**  For a given dataset, we sort all samples by popularity in ascending order, remove duplicates, and divide the popularity values into 10 equal intervals, assigning values from 1 to 10 in ascending order. Each sample is then assigned to its corresponding interval, updating its popularity accordingly. For 3-shot, we randomly select one sample from the intervals with popularity values of 2, 5, and 8. For 5-shot, we randomly select one sample from the intervals with popularity values of 1, 3, 5, 7, and 9. For 10-shot, we randomly select one sample from each of the 10 intervals.

**Results.**  Figure 7 shows the average prediction accuracy of Conf-QG-R across three datasets using model-generated popularity under different shot settings. As the number of examples increases, prediction accuracy does not improve, while inference cost rises. Therefore, we recommend prompting LLMs to assess their familiarity with entities and their relationships in a zero-shot setting. Due to API costs, we first conduct experiments on LLaMA3-8B-Instruct and Qwen2-7B-Instruct and find that increasing the number of samples in the prompt does not yield more effective knowledge popularity. Therefore, we only perform 0-shot and 3-shot experiments on ChatGPT.

## D  Detailed Parameter Settings

**Inference.**  For all the models, we use greedy search, selecting the token with the highest probability at each generation step. For open-source models, our experiments are conducted on a single 80GB A800 GPU.

**MLP Training.**  For both corpora-based and model-generated popularity, we train the model using the Adam optimizer with a learning rate of 2e-3 and a batch size of 8. The intermediate layer has a dropout rate of 0.4, and training runs for 100 epochs. All experiments are conducted on two 16GB V100 GPUs. We select the checkpoint with the highest prediction accuracy on the training set for evaluation on the test set.

**Class Balancing for The Basketball Dataset.** Since the MLP fails to learn meaningful patterns on the basketball dataset for Llama3 and Qwen2—consistently classifying all samples as incorrect due to the overwhelming imbalance—we extract all correctly answered samples and randomly sample an equal number of incorrect ones (seed = 0) to ensure balanced learning across both classes. The training set and the test set are evenly split from the sampled dataset.

# E    CASE STUDIES

We compare PC and PC+ALL on LLaMA3 for answer correctness prediction to illustrate how knowledge popularity works in confidence calibration. The imperfect alignment between the model's confidence and its actual performance arises from two main factors:

- **Overconfidence**: The model generates incorrect answers with high confidence. When classification relies on generation probabilities, such answers are incorrectly labeled as correct.
- **Conservativeness**: The model generates correct answers with low confidence. When classification relies on generation probabilities, such answers are incorrectly labeled as incorrect.

We collect the samples misclassified by PC but successfully calibrated by PC+ALL. These fall into two categories:

- **Overconfidence Group**: Samples where the model generates an incorrect answer, PC incorrectly classifies them as correct, while PC+ALL correctly identifies them as incorrect.
- **Conservativeness Group**: Samples where the model generates a correct answer, PC incorrectly classifies them as incorrect, while PC+ALL correctly identifies them as correct.

We compute the knowledge popularity for each group, and the results appear in Table 6. The results show that in the overconfidence group, PC+ALL achieves calibration by leveraging low knowledge popularity despite the model's high confidence. In contrast, in the conservativeness group, it achieves calibration through high knowledge popularity.

Although PC+ALL achieves strong calibration performance, it also introduces some over-calibration issues by misclassifying samples that were correctly predicted by PC, as shown in Figure 8. However, the number of correctly calibrated samples significantly exceeds the over-calibrated ones. Moreover, we show some cases on the Movies dataset for Llama3. Figures 9 and Figure 10 illustrate cases where knowledge popularity effectively calibrated the model's confidence, while Figure 11 shows a failure case. All the results in this section are obtained with seed=0.

| Datasets | Group | PC | $Pop_Q$ | $Pop_{Ge}$ | $RPop_{Ge}$ |
|---|---|---|---|---|---|
| Movies | Overc. | 0.91 | 47.05 | 20.08 | 1.03 |
| | Conse. | 0.78 | 24.34 | 23.22 | 12.91 |
| Songs | Overc. | 0.91 | 38.91 | 13.04 | 18.89 |
| | Conse. | 0.78 | 131.00 | 15.50 | 103.00 |
| Basketball | Overc. | 0.78 | 102.68 | 10.69 | 0.97 |
| | Conse. | 0.62 | 234.83 | 10.59 | 11.47 |

Table 6: Knowledge popularity of samples that are misclassified by PC but correctly classified by PC+ALL. Overc. refers to the Overconfidence group, in which the model generates an incorrect answer but PC classifies it as correct. Conse. refers to the Conservativeness group, in which the model generates a correct answer but PC classifies it as incorrect.

From Figure 10, we can see that the model generated an incorrect answer with a probabilistic confidence of 0.95, which is significantly higher than the classification threshold for confidence (¿0.85), leading to it being classified as correct. However, knowledge popularity reveals that the question pop, generated answer pop, and relation pop are 16, 20, and 1, respectively, all below the dataset's average levels. This indicates that both the question and the generated entity are relatively uncommon and rarely co-occur. As a result, the classification outcome was corrected to incorrect. Similarly, in Figure 10, the model exhibits low probabilistic confidence for a correctly generated answer, leading to a misclassification as incorrect. However, its knowledge popularity was relatively high, resulting in a correction to the correct classification.

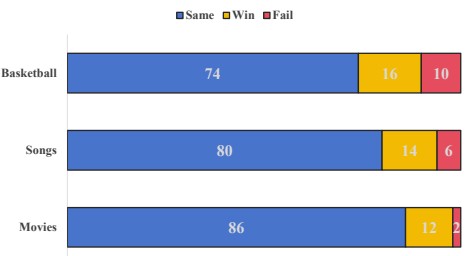

Figure 8: The difference in answer correctness prediction on LLaMA3 between using PC+ALL and using PC. Blue indicates that both methods make the same prediction, yellow indicates cases where only PC+ALL predictes correctly, and red indicates cases where only PC predictes correctly.

**Question**: Who is the director of the movie The Star Maker
**Ground-Truth Answer**: Giuseppe Tornatore
**Generated Answer**: Giuseppe Tornatore
**Correctness**: ✅
**Confidence**: 0.68
**Confidence Threshold**: 0.85
**Correctness Prediction**: ❌
**Knowledge Popularity**: [Q-Pop: 15, G-Pop: 62, R-Pop: 15]
**Average Popularity**: [Q-Pop: 26, G-Pop: 40, R-Pop: 15]
**Correctness Prediction After Calibration**: ✅

Figure 9: The case where Llama3-8B generates incorrect answers with high confidence on the Moveis dataset and is corrected by low knowledge popularity.

**Question**: Who is the director of the movie Itinéraire d'un enfant gâté
**Ground-Truth Answer**: Claude Lelouch
**Generated Answer**: Coline Serreau
**Correctness**: ❌
**Confidence**: 0.95
**Confidence Threshold**: 0.85
**Correctness Prediction**: ✅
**Knowledge Popularity**: [Q-Pop: 16, G-Pop: 20, R-Pop: 1]
**Average Popularity**: [Q-Pop: 26, G-Pop: 40, R-Pop: 15]
**Correctness Prediction After Calibration**: ❌

Figure 10: The case where Llama3-8B generates correct answers with low confidence on the Moveis dataset and is corrected by high knowledge popularity.

**Question**: Who is the director of the movie The Celluloid Closet
**Ground-Truth Answer**: Rob Epstein
**Generated Answer**: Rob Epstein
**Correctness**: ✅
**Confidence**: 0.99
**Confidence Threshold**: 0.85
**Correctness Prediction**: ✅
**Knowledge Popularity**: [Q-Pop: 16, G-Pop: 15, R-Pop: 0]
**Average Popularity**: [Q-Pop: 26, G-Pop: 40, R-Pop: 15]
**Correctness Prediction After Calibration**: ❌

Figure 11: The case where Llama3-8B generates correct answers with high confidence on the Moveis dataset and is misled by low knowledge popularity.

Figure 11 presents a case of error correction. While similar misclassifications may occur, the proportion of correctly corrected samples (6.0%) is significantly higher than that of miscalibrated ones (1.2%), demonstrating the reliability of knowledge popularity in confidence calibration.

## F  PROMPTS

We display all the prompts used in this paper here and show some examples.

**QA prompt.** We just ask the model to give a short answer without any other words. The example is shown in Figure 20.

**Prompts for knowledge popularity generation.** Examples for instructing LLMs to provide question entity popularity, generated answer popularity, and the popularity of their relationship can be found in Figure 22 23 24 25 26 27.

# G THE USE OF LARGE LANGUAGE MODELS

LLMs were used solely for grammar correction and sentence polishing. All content and experiments in this paper were conducted entirely by humans, and any model-polished text was manually reviewed.

| Datasets | Models | Accuracy | | | Confidence | | | Alignment | | |
|---|---|---|---|---|---|---|---|---|---|---|
| | | Q-Pop | G-Pop | Co-Occ | Q-Pop | G-Pop | Co-Occ | Q-Pop | G-Pop | Co-Occ |
| Movies | Llama3-8B | 0.317 | 0.100 | **0.637** | 0.404 | 0.324 | **0.653** | 0.404 | 0.231 | **0.667** |
| | Qwen2-7B | 0.433 | 0.087 | **0.756** | 0.413 | 0.345 | **0.679** | 0.386 | 0.021 | **0.607** |
| | ChatGPT | 0.134 | 0.083 | **0.208** | 0.210 | 0.233 | **0.304** | 0.211 | 0.231 | **0.304** |
| Songs | Llama3-8B | 0.277 | 0.257 | **0.621** | 0.369 | 0.188 | **0.680** | 0.182 | 0.207 | **0.358** |
| | Qwen2-7B | 0.362 | 0.188 | **0.666** | 0.300 | 0.246 | **0.511** | 0.230 | 0.058 | **0.405** |
| | ChatGPT | 0.171 | 0.218 | **0.351** | 0.249 | 0.305 | **0.445** | 0.232 | 0.297 | **0.326** |
| Basketball | Llama3-8B | 0.118 | 0.116 | **0.245** | **0.173** | -0.034 | 0.010 | -0.052 | 0.083 | **0.163** |
| | Qwen2-7B | 0.014 | **0.116** | 0.106 | **0.151** | 0.114 | 0.068 | -0.126 | -0.015 | **0.018** |
| | ChatGPT | 0.288 | -0.164 | **0.293** | **0.351** | -0.210 | 0.257 | 0.201 | -0.107 | **0.241** |

Table 7: Spearman correlation coefficients for Accuracy, Confidence, and Alignment scores with the popularity of question entities, generated entities, and their co-occurrence.

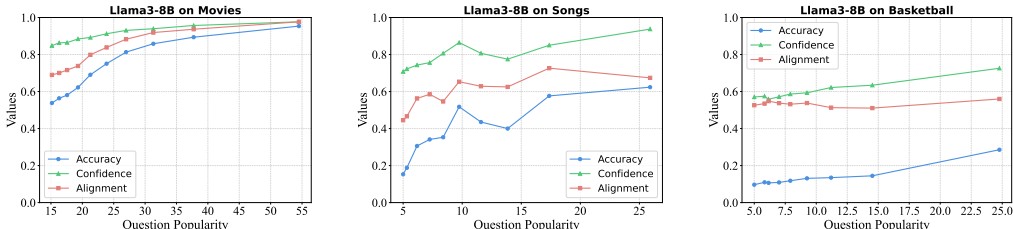

Figure 12: The QA performance, confidence, and alignment of Llama3 under different question popularity.

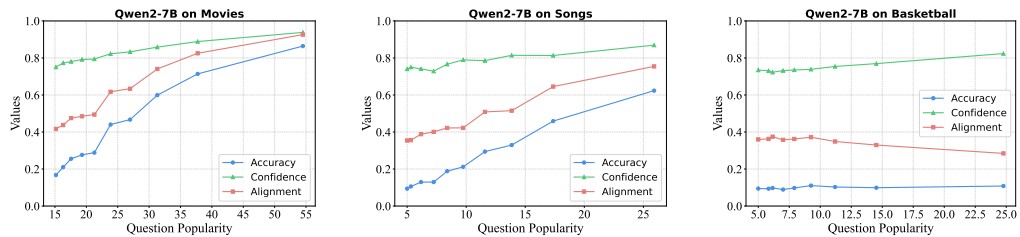

Figure 13: The QA performance, confidence, and alignment of Qwen2 under different question popularity.

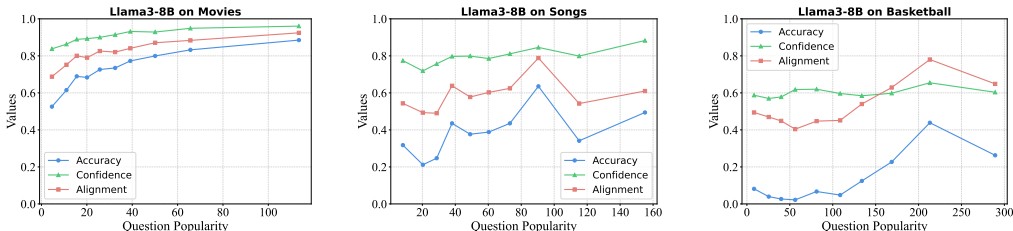

Figure 14: The QA performance, confidence, and alignment of Llama3 under different answer popularity.

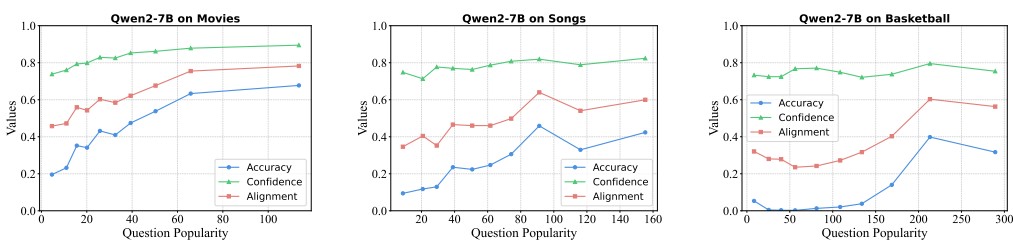

Figure 15: The QA performance, confidence, and alignment of Qwen2 under different answer popularity.

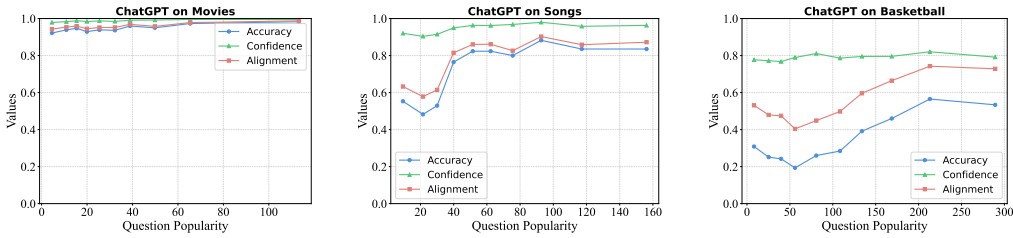

Figure 16: The QA performance, confidence, and alignment of ChatGPT under different answer popularity.

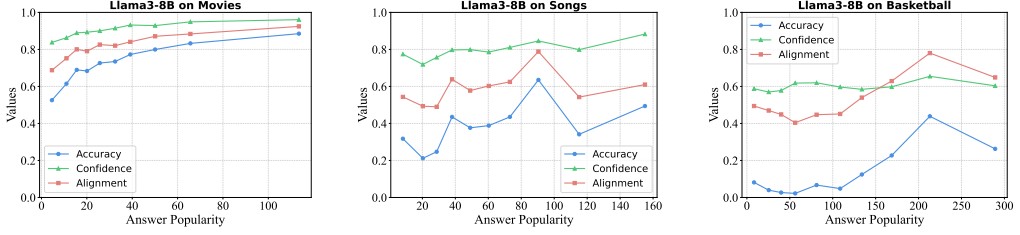

Figure 17: The QA performance, confidence, and alignment of Llama3 under different relation popularity.

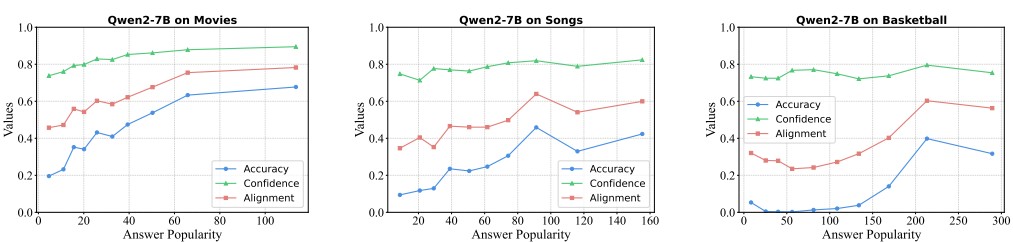

Figure 18: The QA performance, confidence, and alignment of Qwen2 under different relation popularity.

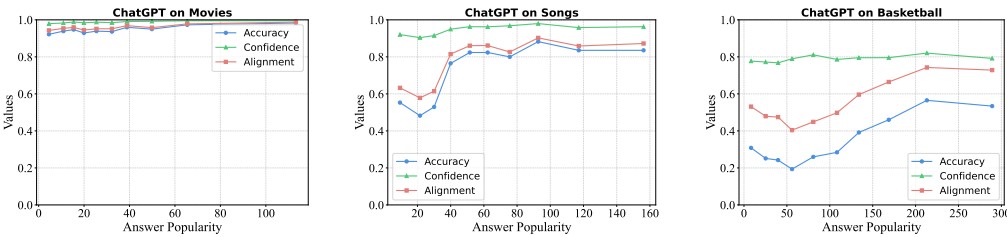

Figure 19: The QA performance, confidence, and alignment of ChatGPT under different relation popularity.

---

**Input**:
  Answer the following question with one or few words.
  Question: Who is the director of the movie The Intouchables
  Answer:
**Response**: Eric Toledano

---

Figure 20: A question-answering example for Llama3.

---

**Input**:
  Judge whether the following answer (this is your self-generated answer) about the question is correct. If you are sure the answer is correct, say certain. If not, please say uncertain. Just give your judgement without any other words.
  **Question**: Where is the birthplace of the basketball player Jiang Xingquan?
  **Answer**: Beijing.
**Response**: Uncertain.

---

Figure 21: An example for verbalized confidence.

---

**Question**: Rate how familiar you are with the **movie** 'The Intouchables'. The familiarity is rated on a scale from 1 to 10, where 10 means you are highly familiar with it, and 1 means you have little to no knowledge about it. Your answer needs to be a precise integer. Provide only the number, without any additional explanation. Number:

**Response**: 8

---

Figure 22: An example of obtaining question popularity on the movies dataset using LLaMA3 in a zero-shot setting.

---

**Question**: Rate how familiar you are with the **director** 'Eric Toledano'. The familiarity is rated on a scale from 1 to 10, where 10 means you are highly familiar with it, and 1 means you have little to no knowledge about it. Your answer needs to be a precise integer. Provide only the number, without any additional explanation. Number:

**Response**: 4

---

Figure 23: An example of obtaining generated answer popularity on the movies dataset using LLaMA3 in a zero-shot setting.

**Question**: Rate how familiar you are with the **relationship** between the **movie** 'The Intouchables' and the **director** 'Eric Toledano'. The familiarity is rated on a scale from 1 to 10, where 10 means you are highly familiar with their relationship, and 1 means you know little to nothing about it. Your answer needs to be a precise integer. Provide only the number, without any additional explanation.
Number:

**Response**: 8

Figure 24: An example of obtaining relation popularity on the movies dataset using LLaMA3 in a zero-shot setting.

**Question**: Rate how familiar you are with the **movie** 'Swept Away'. The familiarity is rated on a scale from 1 to 10, where 10 means you are highly familiar with it, and 1 means you have little to no knowledge about it. Your answer needs to be a precise integer. Provide only the number, without any additional explanation.
**Here are some examples**:
The movie: Matchstick Men
Number: 2
The movie: Kick-Ass
Number: 5
The movie: Skyfall
Number: 8
Rate how familiar you are with the **movie** 'Swept Away'. The familiarity is rated on a scale from 1 to 10, where 10 means you are highly familiar with it, and 1 means you have little to no knowledge about it. Your answer needs to be a precise integer. Provide only the number, without any additional explanation.
Number:

**Response**: 3

Figure 25: An example of obtaining question popularity on the movies dataset using ChatGPT in a 3-shot setting.

**Question**: Rate how familiar you are with the **director** 'Guy Ritchie'. The familiarity is rated on a scale from 1 to 10, where 10 means you are highly familiar with it, and 1 means you have little to no knowledge about it. Your answer needs to be a precise integer. Provide only the number, without any additional explanation.
**Here are some examples**:
The director: James McTeigue
Number: 2
The director: Guy Ritchie
Number: 5
The director: Jodie Foster
Number: 8
Rate how familiar you are with the **director** 'Guy Ritchie'. The familiarity is rated on a scale from 1 to 10, where 10 means you are highly familiar with it, and 1 means you have little to no knowledge about it. Your answer needs to be a precise integer. Provide only the number, without any additional explanation.
Number:

**Response**: 7

Figure 26: An example of obtaining answer popularity on the movies dataset using ChatGPT in a 3-shot setting.

**Question**: Rate how familiar you are with the relationship between the **movie** 'Swept Away' and the **director** 'Guy Ritchie'. The familiarity is rated on a scale from 1 to 10, where 10 means you are highly familiar with their relationship, and 1 means you know little to nothing about it. Your answer needs to be a precise integer. Provide only the number, without any additional explanation.
**Here are some examples**:
The movie: Kick-Ass; The director: Matthew Vaughn
Number: 2
The movie: Eraserhead; The director: David Lynch
Number: 5
The movie: Heat; The director: Michael Mann
Number: 8
Rate how familiar you are with the relationship between the **movie** 'Swept Away' and the **director** 'Guy Ritchie'. The familiarity is rated on a scale from 1 to 10, where 10 means you are highly familiar with their relationship, and 1 means you know little to nothing about it. Your answer needs to be a precise integer. Provide only the number, without any additional explanation.
Number:

**Response**: 7

Figure 27: An example of obtaining relation popularity on the movies dataset using ChatGPT in a 3-shot setting.

