# OpenReview forum: "How Knowledge Popularity Influences and Enhances LLM Knowledge Boundary Perception"
_ICLR.cc/2026/Conference — ICLR 2026 Conference Withdrawn Submission_

### Official Review · Reviewer_K9bt · 2025-10-24

**Soundness:** 2
**Presentation:** 2
**Contribution:** 2
**Rating:** 2
**Confidence:** 5

**Summary:**

This work examines the relationship between the popularity of specific knowledge and an LLM’s perception of its own knowledge boundaries. The authors quantify popularity using three distinct metrics: 1) the popularity of the entity found in the question ($Pop_Q$), 2) the popularity of the entity in the correct answer ($Pop_{GT}$), and 3) the popularity of the relation, defined by the co-occurrence of the question and ground-truth entities ($RPop_{GT}$).

**Strengths:**

- The study methodically measures knowledge popularity through three distinct lenses: the entity within the question, the entity within the answer, and the co-occurrence (relation) of both. This enables an examination of how popularity influences the performance and confidence of LLMs.
- Leveraging these popularity metrics as signals, the research enhances answer correctness prediction accuracy by an average of 5.24% across all evaluated models and datasets.
- It also proposes a feasible technique for LLMs to assess popularity independently, which mitigates the dependency on external data sources and the expenses associated with their collection.

**Weaknesses:**

- A significant practical limitation of the proposed calibration approach is its dependence on the Probabilistic Confidence metric. While PC was not introduced in this paper, the entire analysis and the resulting calibration technique are fundamentally restricted to models that allow users access to token probabilities. This dependency is a major obstacle for real-world scenarios involving black-box API-based LLMs. The paper should explicitly mention this limitation.
- Another critical point is that the paper fails to discuss highly relevant existing work, such as [1] and [2]. The omission of [2] is particularly concerning, as that study also quantified popularity by counting entities and entity-relation pairs from a Wikipedia dump, a methodology very similar to the one employed in this paper. A comparative discussion with these papers is required to properly situate this work and clarify its contributions.
- The analytical approach of using simple entity co-occurrence as a proxy for “relation popularity” remains a key concern. This definition ignores the specific semantics of the relationship. It seems counterintuitive that different relations (e.g., birthplace vs team location) would share an identical popularity score just because they involve the same entity pair.
The authors demonstrate a correlation for the three specific relations tested, but this finding may not be generalizable, i.e., other relations may follow different trends. To substantiate the claim, further investigations are needed. For example, the authors should conduct a comparative analysis involving the same entity types but different relations, such as contrasting (baseball player, birthplace, location) with (baseball player, plays in, location).

[1] Sun, Kai, et al. "Head-to-tail: how knowledgeable are large language models (LLMs)? AKA will LLMs replace knowledge graphs?." NAACL 2024.

[2] Maekawa, Seiji, et al. "Retrieval Helps or Hurts? A Deeper Dive into the Efficacy of Retrieval Augmentation to Language Models." NAACL 2024.

Minor points:

- Lines 191 and 192 seem to include undefined characters, such as inverted question marks and exclamation marks. What do these symbols signify?
- Line 499 includes Section ??. Please fix it.

**Questions:**

Could you explain all the points I raised in the Weaknesses section?

---

### Official Review · Reviewer_NELu · 2025-10-27

**Soundness:** 2
**Presentation:** 2
**Contribution:** 2
**Rating:** 4
**Confidence:** 4

**Summary:**

The paper studies the relationship between knowledge popularity and a model’s confidence in answering questions regarding that knowledge. Accuracy, confidence, and the alignment between the two are investigated, as experiments are carried out on entity-centric QA.

The authors find that models deliver stronger accuracy, confidence, and alignment on more popular knowledge. An MLP-based calibration technique is proposed to utilize the strong correlation with relation popularity.

**Strengths:**

- The idea to explore the alignment of model confidence and its actual performance is intriguing.
- The paper offers an interesting finding regarding model hallucination: the incorrect answers often have better popularity than ground truth answers.
- Popularity-based calibration is simple and effective in improving model accuracy.

**Weaknesses:**

- The paper fails to acknowledge an important prior work by Kandpal et al. [1], which first studied the relationship between co-occurrence frequency and a model’s factual accuracy, and demonstrated the correlation and causation between the two.
- The analysis is mostly correlational. No causal interventions are taken to prove it is indeed popularity that causes the variation in accuracy and confidence.
- The MLP's simple structure likely makes it prone to performing poorly on imbalanced training data, as mentioned in Appendix D.
- The findings would be more compelling if tested on the newer models.

[1] Large Language Models Struggle To Learn Long-Tail Knowledge (Kandpal et al., 2022)

**Questions:**

- Do you train a separate MLP for every dataset? Or is there only one generalist MLP for all datasets?

---

### Official Review · Reviewer_ktQm · 2025-10-31

**Soundness:** 4
**Presentation:** 3
**Contribution:** 2
**Rating:** 4
**Confidence:** 4

**Summary:**

Summary:

 - Motivation: LLMs often fail to recognize their knowledge boundaries.
 - This paper focuses on how knowledge popularity affects LLMs’ ability to perceive their knowledge boundaries.
 - Specifically they focus on entity-centric factual question answering (QA) and quantify it in three perspectives: (a) the popularity of entities in the question, (b) the popularity of entities in the answer, and (c) relation popularity, defined as their co-occurrence frequency. Specifically they release PopQ, PopGT, and RPopGT for future research.

 - The experiments show that, LLM performance correlates strognly with knowledge popularity, with relation popularity having the strongest correlation.
 - They also look into prompting LLMs to estimate popularity (without external corpora).

**Strengths:**

Overall, while the overall findings of this work is quite predictable (we have known them from Mallen et al. and other related works) I tend to like this work as it extends the prior datasets along domains which will be super-useful for the field. Despite my positivity, one may argue that this paper should be published in a resource-centric venue which may be a fair argument.

**Weaknesses:**

Like I said in the previous response, I think the main weakness of this work is that the overall findings of this work is quite predictable (we have known them from Mallen et al. and other related works).
So the main sales pitch is additional resources.
Is that enough to warrant an ICLR paper? Unsure but I am leaning to say that it's not enough.

**Questions:**

Are these formatings intentions?
 - "Movies ¿ Songs ¿ Basketball"
 - "Movies ¡ Songs ¡ Basketbal"

---

### Official Review · Reviewer_LCnS · 2025-10-31

**Soundness:** 3
**Presentation:** 2
**Contribution:** 3
**Rating:** 6
**Confidence:** 4

**Summary:**

The paper presents a study on how entity knowledge popularity correlates with QA performance, model confidence, and calibration accuracy (if the LM confidence matches the downstream performance). Results show that QA performance, confidence and calibration accuracy increase with more popular entities. They look at the popularity of (1) the question entity, (2) the ground truth / predicted answer entity, and (3) the relation between the question and the ground truth / predicted answer entity.  The popularity of the relation between the generated answer and the question entity correlates with these numbers more strongly. They use entity popularities as features for confidence calibration, and find that they provide some gains on top of LM generation probability.

**Strengths:**

1. The experimental setup is clear and easy to understand.
2. The analysis of correlations between knowledge popularity and model confidence is novel. There are some interesting insights.
3. There is potential in using popularity as a feature for confidence calibration.

**Weaknesses:**

1. The baselines for confidence calibration are pretty weak. For example, Kadavath et al., 2022 seems like a good candidate on top of PC? Also, since you are training the MLPs, I think you should consider implementing some baselines in paragraph “Confidence Estimation via LLM Internal States.” in related work.
2. The presentation of analysis can be a bit hard to follow. It is unclear what are the most important takeaways. Maybe it would be good to summarize important findings in the conclusion. It could also be useful if there are bullet points listing findings in the intro.
3. The analyses are interesting, but I am not sure if there is something that is directly useful. Some of the findings are obvious like L250: “LLMs achieve better QA performance …”. This happens a lot of times for analysis papers, but I think this will make the paper weaker. I like how you use these as methods for confidence calibration, but I am not sure if this is a more useful feature than models’ hidden states (given you’re also training an MLP).
4. The results seem to depend a lot on the data distribution (how popular are the entities), and may not generalize to other domains. For example in Table 3, the results are pretty different between Basketball and the other domains.
5. The gains for confidence calibration without access to the corpus are pretty marginal. It is not very realistic to have a corpus to estimate the popularity, and I think generating popularity using LMs is more practical. However, the results are not as promising.

Reference:
Kadavath, Saurav, et al. "Language Models (Mostly) Know What They Know." CoRR (2022).

**Questions:**

1. In L134-L135, there are claims that “probabilistic confidence has been shown to perform well”. But there are also claims that LMs are overconfident? How are the two both true?
2. Could you also show the variance in Table 5? Also are there statistical significance measures?
3. There are typos in L190-L192. I am not sure about the popularity ranks.
4. How did you decide on these domains? What are the average popularity numbers for these domains? I think it is shown in Yuksekgonul et al., 2023?
5. Have you ever experimented with other correlation metrics? For example, why not Pearson correlation?
6. Table 4 shows that  RPop_Ge is a lot more correlated than the others. Is it possible that this is a dataset artifact? That most questions ask about more popular relations. The result basically shows if the generated answer co-occurs more with the question entity, then it is more likely correct. However, if you intentionally query the model with rare relations that are only seen once in the corpus, this correlation would not hold. When we test LMs capabilities (as they are stronger now), we increasingly turn to the long-tail of the distribution, and the case that I mention will become more likely.
7. There are other metrics for confidence calibration (e.g. Expected Calibration Error), and I wonder why you arrive on the most simple metric, which is “answer correctness prediction accuracy”?


Reference:
Yuksekgonul, Mert, et al. "Attention Satisfies: A Constraint-Satisfaction Lens on Factual Errors of Language Models." The Twelfth International Conference on Learning Representations.

---

### Note · Authors · 2025-12-03

**Comment:**

Thank you to all reviewers for reviewing our paper and for your valuable suggestions. We are continuing to improve the quality of the manuscript.

**Withdrawal Confirmation:**

I have read and agree with the venue's withdrawal policy on behalf of myself and my co-authors.